# Stevioside Enhances the Anti-Adipogenic Effect and β-Oxidation by Activating AMPK in 3T3-L1 Cells and Epididymal Adipose Tissues of *db/db* Mice

**DOI:** 10.3390/cells11071076

**Published:** 2022-03-23

**Authors:** Miey Park, Hana Baek, Jin-Young Han, Hae-Jeung Lee

**Affiliations:** Department of Food and Nutrition, Institute for Aging and Clinical Nutrition Research, College of BioNano Technology, Gachon University, Seongnam 13120, Korea; mpark@gachon.ac.kr (M.P.); ruru456123@naver.com (H.B.); mongmong072@naver.com (J.-Y.H.)

**Keywords:** stevioside, adipogenesis, 3T3-L1 preadipocytes, differentiation, *db/db* mice

## Abstract

Stevioside, the primary sweetener in stevia, is a glycoside with numerous beneficial biological activities. However, its anti-adipogenic effects on tissue differentiation and adipose tissues remain to be thoroughly investigated. In this study, the anti-adipogenic effects of stevioside during the differentiation of 3T3-L1 cells and epididymal adipose tissues of *db/db* mice were investigated by measuring the lipid droplets stained with Oil Red O and an immunoblot assay. Immunoblot analysis revealed that stevioside downregulated the expression of peroxisome proliferator-activated receptor-gamma (PPARγ), sterol regulatory element-binding protein-1c (SREBP-1c), CCAAT/enhancer-binding protein alpha (C/EBPα), and fatty acid synthase (FAS). Additionally, the protein expression of carnitine palmitoyltransferase 1 (CPT1), silent mating type information regulation 2 homolog 1 (SIRT1), and peroxisome proliferator-activated receptor gamma coactivator 1 alpha (PGC-1α) increased following treatment with stevioside. Furthermore, stevioside increased the phosphorylation of adenosine monophosphate (AMP)-activated protein kinase (AMPK) and acetyl-CoA carboxylase (ACC), both in vitro and in vivo. The activity of AMPK in stevioside-treated 3T3-L1 cells was further confirmed using agonists and antagonists of AMPK signaling. Our data indicate that stevioside ameliorates anti-adipogenic effects and promotes β-oxidation in adipocytes by activating AMPK-mediated signaling. The results of this study clearly demonstrated the inhibitory effect of stevioside on the differentiation of adipocytes and the reduction of lipid accumulation in the epididymal adipose tissues of *db/db* mice.

## 1. Introduction

Increasing affluence has various side effects, accompanied by the increasing emergence of overweightness and obesity, the harmful effects of which are quantified to exceed those of alcohol abuse or smoking [1,2]. Various obesity-associated comorbidities are officially recognized, including cardiovascular diseases, certain types of cancer, and diabetes [3,4]. Additionally, increased sugar intake contributes to the overall energy density of the diet, thereby promoting obesity, and the consumption of sugary beverages is associated with complications of obesity-induced heart disease [5,6,7]. Experimental evidence has demonstrated that the main component of sugar and high-fructose corn syrup is fructose, which is responsible for metabolic syndrome [8,9,10]. Fructose may have a significant role in metabolic syndrome regardless of weight gain, inducing of fatty liver, insulin resistance, and hypertension, even when excessive calorie intake is controlled [9,11]. Previous studies have reported that fructose reduces the intracellular levels of ATP and mitochondrial oxidative stress, thereby leading to increased lipogenesis and inhibition of fatty acid (FA) oxidation [12,13].

Obesity is associated with impaired FA oxidation in the liver, muscles, and adipose tissues [14]. Carnitine palmitoyltransferase 1 (CPT1) in the outer mitochondrial membrane enables β-oxidation by converting long-chain acyl-CoA into long-chain acylcarnitine. CPT1 is regulated by a ligand-activated transcription factor, peroxisome proliferator-activated receptor-alpha (PPARα) [15]. PPARα is a nuclear receptor that suppresses inflammatory reactions and lipid peroxidation in the liver and adipose tissues [16]. It has been demonstrated that silent information regulator 1 (SIRT1) also binds to PPARα and is suppressed by the adipose tissue regulator, PPARγ, in white adipocytes [17]. The lack of adenosine monophosphate (AMP)-activated protein kinase (AMPK) in adipocytes serves as a crucial energy sensor and induces obesity in response to nutrient overload and related metabolic dysfunction [18]. The activation of AMPK inhibits FA synthesis and promotes FA oxidation by phosphorylating acetyl-CoA carboxylase (ACC). The activity of CPT1 is therefore restored by the activation of AMPK [19].

*Stevia rebaudiana Bertoni* (SB) is a low-calorie sweetener that is 300 times sweeter than saccharose and contains folic acid, vitamin C, and essential amino acids [20,21]. The single main ingredient that accounts for 4% to 20% of the dry weight of the leaves of SB is the deter-fen glycoside, stevioside [22]. Both the SB plant and stevioside are safe for use as sweeteners, are calorie-free, and suitable for people with diabetes and obesity. They have been used for several years in many countries worldwide [23,24]. Sufficient evidence suggests that stevioside has anti-inflammatory effects, inhibits tumor progression in skin cancer, and enhances innate immunity [25,26,27]. In addition, the dietary supplementation of stevioside alleviates intestinal mucosal damage and protects against myocardial fibrosis [28,29]. However, different research groups continue to hold dissimilar opinions regarding the efficacy of stevia against diabetes [30,31]. Additionally, there is no clear evidence to suggest that stevioside decreases lipid formation and promotes lipid metabolism by activating AMPK-mediated signals in adipocytes in *db/db* mice. Therefore, the purpose of this study was to determine whether stevioside exerts anti-adipogenic effects on AMPK-mediated signaling in 3T3-L1 cells and *db/db* mice.

## 2. Materials and Methods

### 2.1. Materials

Stevioside (C38H60O18; MW 804.87) was purchased from Sigma-Aldrich (St. Louis, MO, USA). Insulin was procured from Thermo Fisher Scientific (San Jose, CA, USA), while 3-isobutyl-1-methylxanthine (IBMX), dexamethasone (DEX), aminoimidazole-4-carboxamide ribonucleotide (AICAR), and dorsomorphin (Compound C) were purchased from Sigma-Aldrich (St. Louis, MO, USA).

### 2.2. Cell Culture and Differentiation of Pre-Adipocytes

For the in vitro experiments, 3T3-L1 pre-adipocytes were obtained from the American Type Culture Collection (ATCC; Manassas, VA, USA) and maintained in Dulbecco’s Modified Essential Medium (DMEM) supplemented with 10% bovine calf serum (BCS) and antibiotic-antimycotic solution (Thermo Fisher, San Jose, CA, USA) in a 5% CO2 incubator at 37 °C. The differentiation of pre-adipocytes was initiated by treating the cells with a differentiation cocktail medium (5 μg/mL insulin, 0.5 mΜ IBMX, and 1 μΜ DEX in DMEM supplemented with 10% FBS) for 3 days. After 3 days of induction, the cells were placed in a post-differentiation medium (10% FBS and 5 μg/mL insulin) for more than 4 days, and the medium was changed every 2 days. On day 7, the completely differentiated white adipocytes were used for further experiments. Stevioside was treated by concentration from the beginning of the differentiation.

### 2.3. Quantification of Lipids Droplets

Following differentiation, the control and stevioside-treated cells were rinsed with PBS and fixed with 4% paraformaldehyde for more than 1 h. The cells were subsequently washed with 60% isopropanol and dried. Each well was stained with filtered Oil Red O working solution for 1 h at room temperature and washed thrice. Images of the stained lipid droplets in the cells were captured using a microscope (Nikon Eclipse, Shinagawa, Tokyo, Japan). After capturing the images, the stained lipid droplets were eluted with 100% isopropanol, and the absorbance was measured at 500 nm.

### 2.4. Animals and Diets

Eight-week-old male *db/db* mice (BKS.Cg-Dock7m +/+ *Leprdb*/J, homozygote) and age-matched negative controls (C57BL/6J mice, heterozygotes) were purchased from Jackson Laboratories (Sacramento, CA, USA). All the mice were maintained in a controlled environment, at a temperature of 20–25 °C, relative humidity of 60%, under a 12 h/12 h light-dark cycle, and were given ad libitum access to food (normal rodent diet). After 2 weeks of acclimatization, the mice were randomly divided into 3 groups, namely, the negative control group that received saline (C57BL/6), a control group that received saline (*db/db*), and a control group that received 40 mg/kg of stevioside (*db/db*_SS). Saline and stevioside dissolved in saline were administered to the mice once daily by oral gavage for 3 weeks. All mice were starved for 12 ± 1 h prior to sacrifice. All the animal experiments were performed in accordance with the *Guide for the Care for and Use of Laboratory Animals* and approved by Gachon University (GIACUC-R2020012).

### 2.5. Histological Analysis of Epididymal Tissue

For paraffin embedding, the epididymal adipose tissues were fixed in 10% formalin (Sigma-Aldrich, St. Louis, MO, USA). The paraffin sections (3–4 μM thick) were stained with hematoxylin and eosin, following which images were captured using a Nikon DS-Ri2 camera (Nikon, Tokyo, Japan), and quantified using ImageJ software (version 1.8.0), rsb.info.nih.gov/ij (accessed on 5 August 2021).

### 2.6. Immunoblot Analysis

Proteins were extracted from 3T3-L1 cells and samples of epididymal adipose tissues using a protein lysis buffer (iNtRON Biotechnology, Seongnam, Korea) with phosphatase and protease inhibitors. The protein samples (30 μg) were separated by sodium dodecyl sulfate-polyacrylamide gel electrophoresis, and the separated bands were transferred to a PVDF membrane, and subsequently blocked in 5% skim milk. The membranes were incubated overnight with the primary antibodies for 2 h at room temperature (RT) or 4 °C. The membranes were subsequently incubated with horseradish peroxidase-labeled secondary antibodies for 1 h. The reactive bands of target proteins were detected using an ECL system (iNtRON Biotechnology, Seongnam, Korea), and the reactive band signals were visualized using a Quant LAS 500 system (GE Healthcare Bio-Sciences, Björkgatan, Uppsala, Sweden).

### 2.7. Statistical Analyses

All the results are expressed as the mean ± standard deviation (SD). The experiments were performed thrice, and each experiment was performed in triplicate. Statistical analysis was performed using GraphPad Prism, version 9.02 (GraphPad Software Inc., La Jolla, CA, USA) with one-way ANOVA and Tukey’s post hoc test. The differences were considered to be statistically significant when the *p*-value was less than 0.05.

## 3. Results

### 3.1. Effects of Stevioside on Lipid Accumulation in 3T3-L1 Adipocytes

The cytotoxicity of stevioside on 3T3-L1 pre-adipocytes was also investigated in this study. No cytotoxicity was observed when the cells were treated with stevioside at concentrations within 200 μM, for 24 or 72 h (Appendix A). The 3T3-L1 pre-adipocytes were allowed to differentiate for seven days, with different concentrations of stevioside. The cells were subsequently stained with Oil Red O for studying lipid accumulation in 3T3-L1 adipocytes. The intracellular lipids in stevioside-treated cells decreased in a dose-dependent manner following treatment with stevioside, compared to those of the cells in the negative control (Figure 1).

### 3.2. Effects of Stevioside on the Expression of Adipogenic Proteins in 3T3-L1 Adipocytes

In order to investigate the effects of stevioside on adipogenesis, we performed immunoblot analysis to determine the protein expression of PPARγ, sterol regulatory element-binding protein-1 (SREBP-1), CCAAT/enhancer-binding protein alpha (C/EBPα), and FA synthase (FAS). As depicted in Figure 2, stevioside significantly reduced the protein expression of PPARγ, SREBP-1, C/EBPα, and FAS in a dose-dependent manner.

### 3.3. Effects of Stevioside on the Expression of Proteins Related to FA Oxidation in 3T3-L1 Adipocytes

We examined the effects of stevioside on FA oxidation in 3T3-L1 adipocytes by studying the protein expression of CPT1 and SIRT1. As depicted in Figure 3, stevioside increased the protein levels of CPT1 (Figure 3a,d) and SIRT1 (Figure 3b,d) at high concentrations of 100 and 200 μM, compared to those of the control groups that received 0 μM stevioside. Moreover, stevioside increased the highest level of PGC-1α (Figure 3c,d) expression in 100 μM.

### 3.4. Effects of Stevioside on the Phosphorylation of AMPK and ACC in 3T3-L1 Adipocytes

We next examined the phosphorylation of AMPK and ACC using Western blotting in 3T3-L1 adipocytes. As depicted in Figure 4, stevioside significantly increased the expression of activated AMPK (*p*-AMPK/AMPK) and ACC (*p*-ACC/ACC). The relative expression of *p*-AMPK/AMPK and *p*-ACC/ACC following treatment with 100 μΜ stevioside was significantly upregulated compared to that of the control groups that received 0 μM stevioside.

### 3.5. Effects of Stevioside on the OGTT and ITT in db/db Mice

In order to determine whether stevioside remains active in vivo, stevioside was administered orally at a dose of 40 mg/kg/day to *db/db* mice for 3 weeks. Oral administration of stevioside showed improved glucose resistance compared to *db/db* mice (Figure 5a,b) and also showed improved results in insulin resistance (Figure 5c,d). In serum TG and TC results, *db/db* mice showed higher levels than C57BL/6, but the results of *db/db*_SS were significantly decreased (Figure 5e,f).

### 3.6. Effects of Stevioside on the Epididymal Adipose Tissues of db/db Mice

The size of epididymal adipocytes in the *db/db* group increased significantly in comparison to that in the C57BL/6 group but decreased in size following the oral administration of stevioside. However, the weight of epididymal fat in *db/db*_SS was decreased, but it was not significant (Figure 6).

### 3.7. Effects of Stevioside on Protein Expression in the Epididymal Adipose Tissue

In order to evaluate the effects of stevioside on epididymal adipose tissues, we examined the expression levels of proteins related to adipogenesis and FA oxidation in epididymal adipose tissues. The protein levels of PPARγ, SREBP-1c, C/EBPα, and FAS increased in the *db/db* group compared to those in the C57BL/6J group. However, the increased protein levels decreased significantly in the *db/db*_SS group. Additionally, the protein expressions of CPT1, SIRT1, and PGC-1α increased considerably in the *db/db*_SS group (Figure 7).

### 3.8. Effects of Stevioside on the Phosphorylation of AMPK and ACC in Epididymal Adipose Tissue

The *db/db* mice were orally administered stevioside for 3 weeks. The levels of phosphorylated AMPK (*p*-AMPK/AMPK) and ACC (*p*-ACC/ACC) in the epididymal adipose tissues of *db/db* mice were lower than those in the C57BL/6 group. However, the levels of phosphorylated AMPK and ACC significantly increased in *db/db*_SS mice compared to those in the *db/db* group (Figure 8).

### 3.9. Effects of Stevioside on AMPK Signaling in 3T3-L1 Adipocytes

We next determined whether AMPK plays a crucial role in the anti-adipogenic effects of stevioside in 3T3-L1 adipocytes. To this end, we determined the activation of AMPK in 3T3-L1 adipocytes treated with stevioside or co-treated with stevioside and the AMPK agonist (AICAR, 10 µM), or treated with the AMPK antagonist (dorsomorphin, 10 µM). An immunoblot analysis revealed that stevioside upregulated the expression of *p*-AMPK/AMPK in cells treated with AICAR and dorsomorphin (Figure 9).

## 4. Discussion

Obesity is associated with various metabolic disorders and the excessive accumulation of fat. Obesity is generally caused by the consumption of foods containing a high proportion of fat and sugar and contains more calories than those lost through physical activity. As obesity is an increasing public health problem, numerous studies are being conducted to alleviate obesity-related complications [32]. There are countless anti-obesity treatments; however, they have myriads of side effects. The present study suggests that stevioside significantly reduces lipid accumulation during the differentiation of 3T3-L1 adipocytes by regulating the expression of adipose transcription factors and activating AMPK signaling. We additionally confirmed the effects of stevioside on adipogenesis and FA oxidation in the epididymal adipose tissues of *db/db* mice. The crucial signs of the development of obesity are lipid accumulation and adipocyte differentiation [33].

PPARγ, an essential transcriptional factor of adipogenesis, is a master regulator of lipid storage and also shows increased expression in adipocytes [34]. The activation of PPARγ in adipose tissue converts white adipocytes into brownish adipocytes in vitro [35]. In this study, we found high expression of PPARγ protein in 3T3-L1 adipocytes following treatment with 100 μM stevioside. PPARγ is known to play an important role in adipocyte differentiation and function. This indicated that stevioside could possess a synergistic ability to induce the browning of white fat. It has been demonstrated that along with PPARγ, C/EBPα plays a vital role in regulating the differentiation of mature adipocytes [36]. During the terminal differentiation of adipocytes, C/EBPα stimulates the transcription of SIRT1 expression by directly binding to the SIRT1 promoter [37,38]. SREBP-1c is a transcription factor that has been implicated in the differentiation of adipose tissues and induces the expression of genes involved in the synthesis of glucose and FA [39]. The reduced expression of FAS inhibits the differentiation of adipocytes and lipid accumulation [40]. In our study, immunoblot analysis revealed that stevioside downregulated the protein levels of PPARγ, C/EBPα, SREBP-1c, and FAS in 3T3-L1 cells and in the epididymal adipose tissues of *db/db* mice. In addition, the staining method used for studying lipid accumulation revealed a reduction in lipid droplets in 3T3-L1 adipocytes following treatment with stevioside in a dose-dependent manner, which suggests that stevioside has anti-adipogenic effects.

We also found that stevioside induced PGC-1α protein expression in 3T3-L1 adipocytes and epididymal adipose tissues of *db/db* mice. PGC-1α plays a crucial role in the decision-making of adipocyte cells and, as a metabolic regulator, responds to several stimuli, including nutrients, ROS, hypoxia, insulin, ATP requirements, and cytokines [41]. It plays an essential role in many vital organs, including brown adipose tissue, skeletal muscle, the heart, liver, and kidneys [42]. PGC-1α upregulates the expression of the tricarboxylic acid cycle [43] and has the ability to stimulate peroxisome activity and mitochondrial fatty acid oxidation [44]. In addition, it has a positive effect that may reduce intramuscular lipid deposition and improve tissue insulin sensitivity [42]. One of the PGC-1α dependent myosins is irisin, which is generated by the cleavage of fibronectin type III domain-containing protein 5 (FNDC5) [45]. Irisin stimulates glucose uptake and lipid metabolism through the activation of AMP-activated protein kinase (AMPK) [46,47].

AMPK plays a crucial role in maintaining energy homeostasis in multiple organs and tissues, and enhances AMPK phosphorylation, which leads to lipid breakdown [48,49]. AMPK phosphorylates ACC, a multifunctional enzyme that catalyzes the conversion of acetyl-CoA to malonyl-CoA during lipid synthesis [48,50]. Malonyl-CoA is a potent inhibitor of CPT1 expression. Therefore, an increase in phosphorylated ACC reduces the production of malonyl-CoA in cells and relieves the suppression of CPT1 expression, which in turn increases mitochondrial import and promotes FA oxidation in adipocytes [50,51]. In this study, we demonstrated that stevioside significantly increased the protein expression of phosphorylated AMPK (*p*-AMPK) and reduced the production of ACC, an important enzyme in FA synthesis and decomposition in adipocytes both in vitro and in vivo. In addition, 3T3-L1 cells were treated with an AMPK agonist, AICAR, and an antagonist, dorsomorphin, for further assessment of the stevioside-mediated activation of AMPK. Stevioside enhanced the protein expression of *p*-AMPK/AMPK in the presence of AICAR and dorsomorphin, and these results established that stevioside plays an essential role in activating AMPK.

## 5. Conclusions

This study described the anti-adipogenic effects of stevioside in 3T3-L1 adipocytes and in the epididymal adipose tissues of *db/db* mice by reducing lipid accumulation, downregulating the expression of adipogenic-related proteins, and upregulating the expression of proteins related to FA metabolism. Our findings demonstrate that stevioside increases the phosphorylation of AMPK and ACC in adipocytes both in vitro and in vivo. These results indicate that stevioside could serve as a crucial anti-adipogenic agent in obesity management and provide clear evidence of lipid accumulation in mouse adipocytes.

## Figures and Tables

**Figure 1 cells-11-01076-f001:**
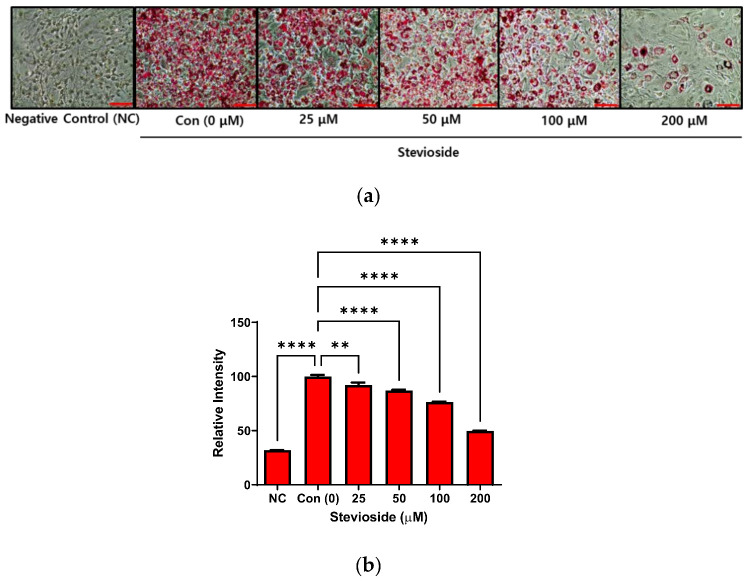
The effect of stevioside on lipid accumulation in 3T3-L1 cells. (**a**) Lipid droplets formation of stained with Oil Red O in 3T3-L1 adipocytes. Scale bar indicates 100 μM. (**b**) The relative intensity of intracellular triglyceride in 3T3-L1 adipocytes. All data are presented as mean ± SD, and tests were performed in three independent experiments. ** *p* < 0.01 and **** *p* < 0.0001. Negative control (NC) was not treated with an adipocyte differentiation cocktail.

**Figure 2 cells-11-01076-f002:**
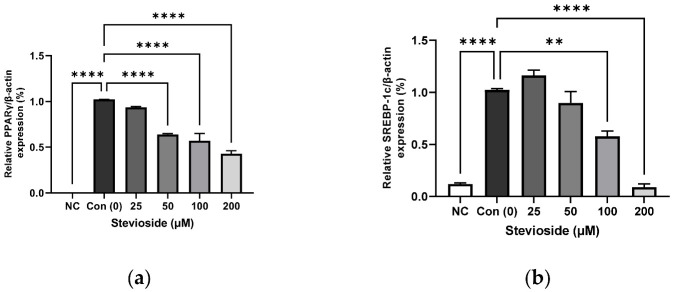
The effect of stevioside on the expression of the adipogenic related proteins in 3T3-L1 cells. Relative protein expressions of (**a**) peroxisome proliferator-activated receptor gamma (PPARγ), (**b**) sterol regulatory element-binding transcription factor-1c (SREBP-1c), (**c**) CCAAT/enhancer-binding protein alpha (C/EBPα), and (**d**) Fatty acid synthase (FAS) were normalized with β-actin. (**e**) Immunoblot analysis of adipogenic related genes in 3T3-L1 cells. All data are presented as mean ± SD, and tests were performed in three independent experiments. ** *p* < 0.01 and **** *p* < 0.0001. Negative control (NC) was not treated with an adipocyte differentiation cocktail.

**Figure 3 cells-11-01076-f003:**
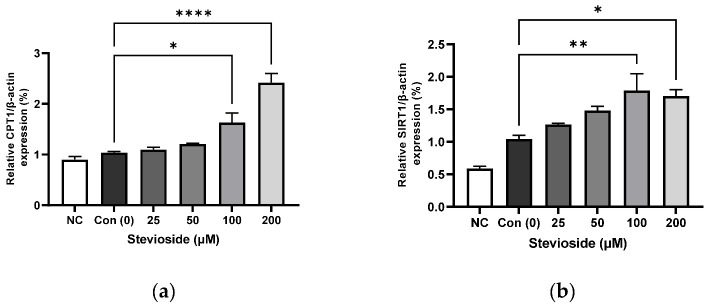
The effect of stevioside on the expression in fatty acid oxidation-related proteins. Relative protein expressions of (**a**) carnitine palmitoyltransferase -1 (CPT-1), (**b**) silent mating type information regulation 2 homolog 1 (SIRT1), and (**c**) peroxisome proliferator-activated receptor gamma coactivator 1 alpha (PGC-1α) were normalized with β-actin. (**d**) Immunoblot analysis of the expression in fatty acid oxidation related protein in 3T3-L1 cells. All data are presented as mean ± SD, and tests were performed in three independent experiments. * *p* < 0.05, ** *p* < 0.01 and **** *p* < 0.0001. Negative control (NC) was not treated with an adipocyte differentiation cocktail.

**Figure 4 cells-11-01076-f004:**
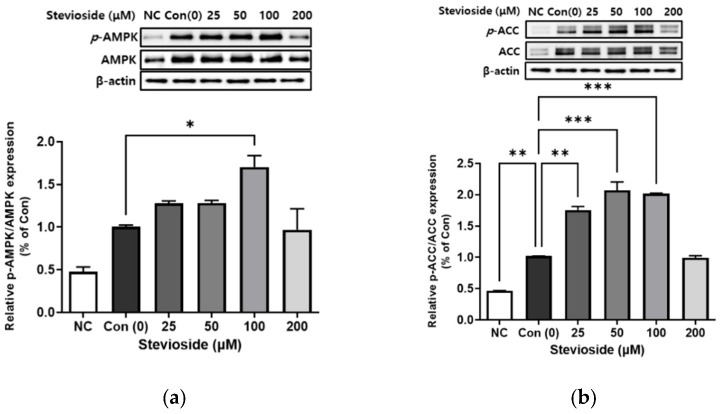
The effect of stevioside on the relative phosphorylation expressions of (**a**) adenosine monophosphate-activated protein kinase (AMPK) (**b**) acetyl-CoA carboxylase (ACC) in 3T3-L1 cells. All data are presented as mean ± SD, and tests were performed in three independent experiments. * *p* < 0.05, ** *p* < 0.01 and *** *p* < 0.001. Negative control (NC) was not treated with an adipocyte differentiation cocktail.

**Figure 5 cells-11-01076-f005:**
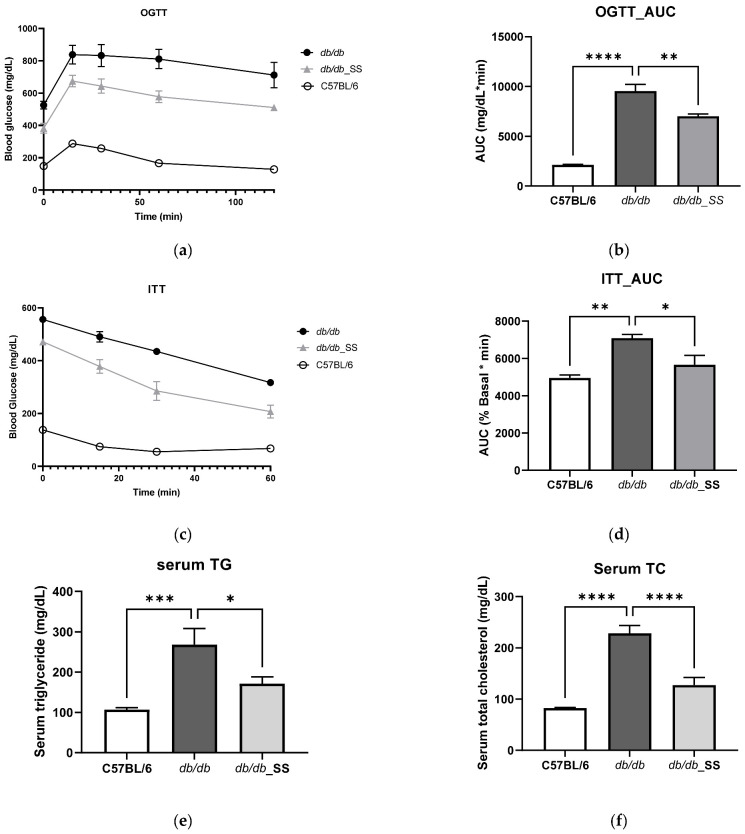
The effect of stevioside on the oral glucose tolerance test (OGTT), insulin tolerance test (ITT), and serum in *db/db* mice. (**a**) OGTT (**b**) OGTT_ area under the curve (AUC) (**c**) ITT (**d**) ITT_AUC (**e**) Serum triglycerides (**f**) Serum total cholesterol. C57BL/6; negative control group that received saline, *db/db*; control group that received saline, *db/db*_SS; control group that received 40 mg/kg of stevioside. All data are presented as mean ± SD, and tests were performed in three independent experiments. * *p* < 0.05, ** *p* < 0.01, *** *p* < 0.001 and **** *p* < 0.0001.

**Figure 6 cells-11-01076-f006:**
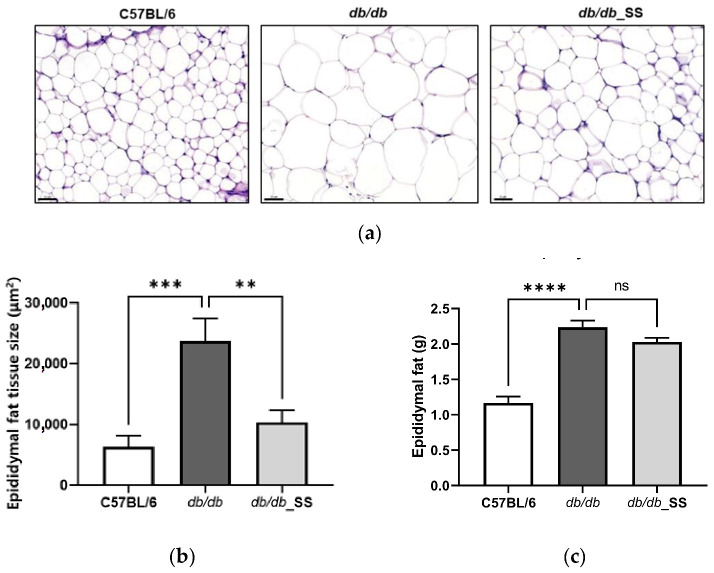
The effect of stevioside on the histological examination of the epididymal adipose tissue of C57BL/6 mice and *db/db* mice. (**a**) Paraffin sections of epididymal adipose tissue of mice treated with stevioside or vehicle were stained with H&E. Scale bar indicates 100 µm. (**b**) The average adipocyte size of epididymal adipose tissue in mice. (**c**) The weight of epididymal fat in mice. C57BL/6; negative control group that received saline, *db/db*; control group that received saline, *db/db*_SS; control group that received 40 mg/kg of stevioside. All data are presented as mean ± SD, and tests were performed in three independent experiments. ** *p* < 0.01, *** *p* < 0.001 and **** *p* < 0.0001.

**Figure 7 cells-11-01076-f007:**
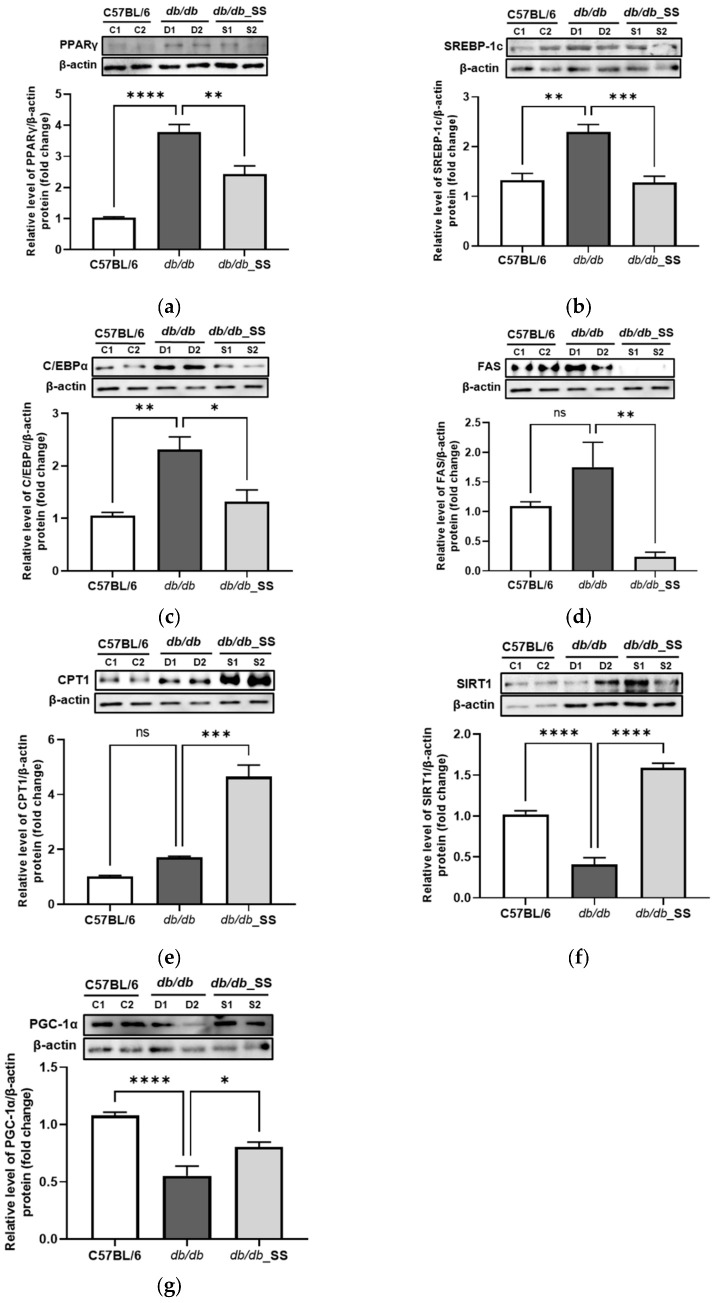
The effect of stevioside on the expression of the adipogenic and fatty acid oxidation related genes in mouse epididymal adipose tissue of C57BL/6 mice and *db/db* mice. (**a**) peroxisome proliferator-activated receptor gamma (PPARγ), (**b**) sterol regulatory element-binding transcription factor-1c (SREBP-1c), (**c**) CCAAT/enhancer-binding protein alpha (C/EBPα), (**d**) fatty acid synthase (FAS), (**e**) carnitine palmitoyltransferase 1 (CPT1), (**f**) silent mating type information regulation 2 homolog 1 (SIRT1) (**g**) peroxisome proliferator-activated receptor gamma coactivator 1 alpha (PGC-1α) were normalized with β-actin. C57BL/6 (C1 and C2); negative control group that received saline, *db/db* (D1 and D2); control group that received saline, *db/db*_SS (S1 and S2); control group that received 40 mg/kg of stevioside. All data are presented as mean ± SD and tests were performed in three independent experiments. * *p* < 0.05, ** *p* < 0.01, *** *p* < 0.001 and **** *p* < 0.0001.

**Figure 8 cells-11-01076-f008:**
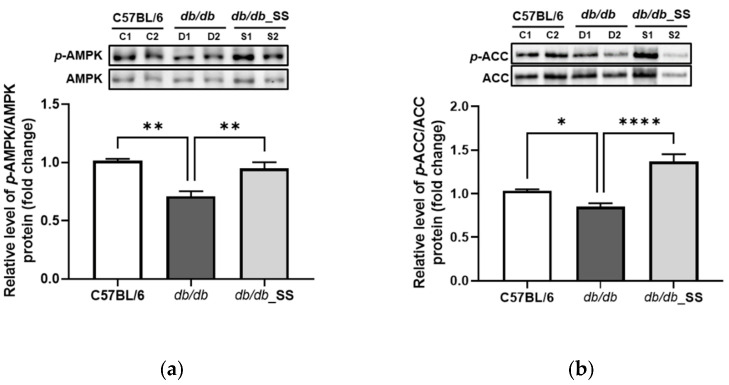
The effect of stevioside on the expression of (**a**) adenosine monophosphate-activated protein kinase (AMPK) and (**b**) acetyl-CoA carboxylase (ACC) in mouse epididymal adipose tissue of C57BL/6 mice and *db/db* mice. C57BL/6 (C1 and C2); negative control group that received saline, *db/db* (D1 and D2); control group that received saline, *db/db*_SS (S1 and S2); control group that received 40 mg/kg of stevioside. data are presented as mean ± SD, and tests were performed in three independent experiments. * *p* < 0.05, ** *p* < 0.01 and **** *p* < 0.0001.

**Figure 9 cells-11-01076-f009:**
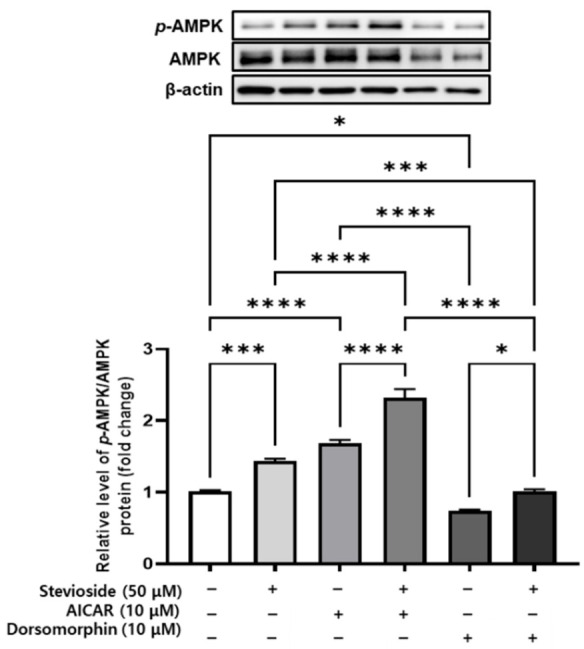
The effect of stevioside on adenosine monophosphate-activated protein kinase (AMPK) activation in AMPK activator-treated and inhibitor-treated in 3T3-L1 cells. *p*-AMPK/AMPK protein expression following stevioside treatment in the presence of 5-aminoimidazole-4-carboxamide ribonucleotide (AICAR; 10 µM) and dorsomorphin (10 µM) in 3T3-L1 cells. All data are presented as mean ± SD, and tests were performed in three independent experiments. * *p* < 0.05, *** *p* < 0.001, and **** *p* < 0.0001.

## Data Availability

Data is contained within the article or Appendix A.

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
