# Peer review of "Stevioside Enhances the Anti-Adipogenic Effect and β-Oxidation by Activating AMPK in 3T3-L1 Cells and Epididymal Adipose Tissues of db/db Mice"

_cells, 2022, doi:10.3390/cells11071076_

Round 1
Reviewer 1 Report
Thanks again for the effects to reviese the manuscript. As the in vivo assay data is included, now I found the revised version mostly satisfactory. But for the final acceptance, there are still two concerns that the authors should address.
- OGTT and ITT procedure should be included in the methods section.
- The Fig.5d (AUC-ITT) and Fig.5c (ITT) are not matched with each other. The curve in db/db SS group is slightly lower than the db/db group but is still clearly higher versus the C57/BL6 control (fig.5c). However, the AUC calculation (fig.5d) shows that these two groups (db/db SS vs control) are almost same. Please correct this error.
Author Response
Reviewer 1
Thanks again for the effects to reviese the manuscript. As the in vivo assay data is included, now I found the revised version mostly satisfactory. But for the final acceptance, there are still two concerns that the authors should address.
Thank you for your encouragement and the opportunity to revise the manuscript. We appreciate your positive feedback on our research and feel that your comments have improved the manuscript considerably. We hope that the revised manuscript will be published in the Cells.
OGTT and ITT procedure should be included in the methods section.
The Fig.5d (AUC-ITT) and Fig.5c (ITT) are not matched with each other. The curve in db/db SS group is slightly lower than the db/db group but is still clearly higher versus the C57/BL6 control (fig.5c). However, the AUC calculation (fig.5d) shows that these two groups (db/db SS vs control) are almost same. Please correct this error.
Thank you for your valuable comment and we are very sorry for the error. We checked our raw data and modified Fig. 5d (AUC-ITT). Kindly refer to the revised manuscript.

Reviewer 2 Report
This manuscript by Park et al. investigated the anti-obesogenic effects of the sweetener stevioside. The authors showed that stevioside treatment of 3T3-L1 adipocytes reduced lipid droplets as well as proteins related to lipogenesis. In addition, they showed that stevioside potentially promotes b-oxidation by activating AMPK signaling. These were also shown in epididymal WAT of db/db mice. This article contains some very interesting ideas; however, several critical procedural and data analysis issues require clarification and re-evaluation, and interpretive issues remain to be addressed. These are all detailed in comments below:
- What is the rationale of using db/db mice instead of diet-induced obese mice? Leptin receptor-deficient db/db mice are commonly used mice models mimicking the conditions of obesity and type 2 diabetes development. However, wouldn’t it be appropriate to use DIO mice in this study if you intend to show the potential of stevioside as a substitute sweetener for obese individuals? Authors mentioned about the deleterious effects of sugar and fructose in the Introduction section. To make the best use of this section, DIO mice seems to be appropriate.
- The schedule of stevioside treatment to 3T3-L1 adipocytes should be described in Materials and Methods section. Was the stevioside added from day 1 of differentiation through the entire differentiation period?
- Which was suppressed in stevioside-treated 3T3-L1 cells, the differentiation or lipid accumulation? Some WAT markers such as Fabp4 and Leptin etc. should be assessed to determine whether lipid accumulation in mature adipocytes is suppressed, or differentiation itself is suppressed.
- In Figure 2e, C/EBPα expression does not seem to be decreased as it is shown in Figure 2c.
- Phosphorylation of ACC leads to suppression of fatty acid synthesis, and it is consistent with the decreased expression of Srebp-1c and Fas. However, it seems to me that the protein expression levels of ACC are increased by stevioside treatment (Figure 4b). Phosphorylated ACC is also increased, but isn’t it because the ACC expression itself is increased by stevioside? The graph does not seem to represent the images. In addition, in the Discussion section, authors wrote that stevioside reduced the production of ACC. This is contradictory.
- The authors aim to demonstrate that β-oxidation is stimulated by stevioside by assessing the expression of CPT1, SIRT1, and PGC-1a. However, the data does not fully support this conclusion. Specifically, authors need to show the expression and phosphorylation levels of lipolytic proteins such as ATGL and HSL, as well as the expression of fatty acid oxidases. Functional demonstration of increased lipolysis such as measurement of NEFA release into the culture medium would also be needed.
- In Figure 6, it is shown that adipocyte size is significantly decreased in db/db_SS mice while the weight of epididymal WAT did not change. Does this mean that the adipocyte number was increased in db/db_SS mice? If so, adipocyte differentiation should be increased by stevioside treatment. Does this phenotype reflect the in vitro results in Figure 1?
- Phosphorylation and expression of AMPK may be affected by the food intake. The authors should state the time they sacrificed the mice, and if they were starved for some time prior to the sacrifice, it should also be stated.
- In Figure 9, stevioside treatment increased the relative level of p-AMPK even in the presence of dorsomorphin. To my understanding, dorsomorphin inhibits the phosphorylation of AMPK, not the expression. So I wonder why stevioside stimulation still phosphorylates AMPK in the presence of dorsomorphin.
- In Discussion section, authors wrote that stevioside could induce the browning of WAT via AMPK signaling. However, this is just a speculation since no experiments had been done to prove this. I think this discussion is not appropriate.
Author Response
Reviewer 2
This manuscript by Park et al. investigated the anti-obesogenic effects of the sweetener stevioside. The authors showed that stevioside treatment of 3T3-L1 adipocytes reduced lipid droplets as well as proteins related to lipogenesis. In addition, they showed that stevioside potentially promotes b-oxidation by activating AMPK signaling. These were also shown in epididymal WAT of db/db mice. This article contains some very interesting ideas; however, several critical procedural and data analysis issues require clarification and re-evaluation, and interpretive issues remain to be addressed. These are all detailed in the comments below:
Thank you for your encouragement and the opportunity to revise the manuscript. We appreciate your positive feedback on our research. We believe that your comments and suggestions have significantly improved the quality of the manuscript. We hope that the revised version will be suitable for publication in Cells.
at is the rationale for using db/db mice instead of diet-induced obese mice? Leptin receptor-deficient db/db mice are commonly used mice models mimicking the conditions of obesity and type 2 diabetes development. However, wouldn’t it be appropriate to use DIO mice in this study if you intend to show the potential of stevioside as a substitute sweetener for obese individuals? The authors mentioned the deleterious effects of sugar and fructose in the Introduction section. To make the best use of this section, DIO mice seem to be appropriate.
à Thank you for your comment. We agree that DIO could be a good choice for this work. However, before the commencement of our research, we thoroughly searched the literature and discovered that many studies have used this mouse model for obesity research. Very humbly, kindly refer to the plausible explanation for your comment.
In our Materials and Methods section, we have described the animal model “Eight-week-old male db/db mice (BKS.Cg-Dock7m +/+ Leprdb/J, homozygote)” which were purchased from Jackson Laboratories (Sacramento, CA, USA). We selected these mice for our study according to their explanation.
They specifically mentioned on their webpage that this strain is ideal for Phase I, II, and III type 2 diabetes research, obesity research, and wound healing.
“Mice homozygous for the diabetes spontaneous mutation (Leprdb) manifest morbid obesity, chronic hyperglycemia, pancreatic beta-cell atrophy, and become hypoinsulinemic. Obesity starts at 3 to 4 weeks of age.”
Please refer to the link below.
https://www.jax.org/strain/000642
https://www.jax.org/news-and-insights/2008/july/bks-cg-m-leprdb-j-000642-one-of-the-most-widely-used-diabetes-models-at-jax
In addition, our study concentrated on the anti-adipogenic effect mainly through in vitro experiments, with in vivo experiments confirming the mechanism. Leptin receptor deficiency is the best candidate tissue researching adipose metabolism. Because of its hypothalamic effects on satiety and energy expenditure, this adipocyte-specific hormone, leptin, which is produced by the obese (ob) gene, regulates adipose-tissue mass. So, we believe that it is difficult to predict that using db/db mice in obesity studies is inappropriate.
According to the experiment, there are various animal models and various induction methods. However, time cannot be overlooked. Also, instead of "obesity", we use "db/db mice" in the title.
db/db mice have been used as an obesity model in several other studies. Please refer to the following literature for further information.
Cruz, Tânia B et al. “Mice with Type 2 Diabetes Present Significant Alterations in Their Tissue Biomechanical Properties and Histological Features.” Biomedicines vol. 10,1 57. 28 Dec. 2021, doi:10.3390/biomedicines10010057
Kämpfer, H et al. “Lack of interferon-gamma production despite the presence of interleukin-18 during cutaneous wound healing.” Molecular medicine (Cambridge, Mass.) vol. 6,12 (2000): 1016-27.
Sahai, Atul, et al. “Obese and diabetic db/db mice develop marked liver fibrosis in a model of nonalcoholic steatohepatitis: role of short-form leptin receptors and osteopontin.” American journal of physiology. Gastrointestinal and liver physiology vol. 287,5 (2004): G1035-43. doi:10.1152/ajpgi.00199.2004
Peng BY, Wang Q, Luo YH, He JF, Tan T, Zhu H. A novel and quick PCR-based method to genotype mice with a leptin receptor mutation (db/db mice). Acta Pharmacol Sin. 2018 Jan;39(1):117-123. doi: 10.1038/aps.2017.52.
The schedule of stevioside treatment to 3T3-L1 adipocytes should be described in the Materials and Memethodson. Was the stevioside added from day 1 of differentiation through the entire differentiation period?
à Thank you for your query. The differentiation schedule has already been described in the materials and methods section (2.2 Cell Culture and Differentiation of Pre-adipocytes). As asked, we added the following sentence to the revised version so that the readers could better understand it.
‘Stevioside was treated by concentration from the beginning of the differentiation’.
Which was suppressed in stevioside-treated 3T3-L1 cells, the differentiation or lipid accumulation? Some WAT markers such as Fabp4 and Leptin etc. should be assessed to determine whether lipid accumulation in mature adipocytes is suppressed, or differentiation itself is suppressed.
à Thank you for your query. The expression of essential genes, including PPARγ, SREBP-1c, and C/EBPα, is required for adipocyte differentiation. Results showed that stevioside inhibits all three expressions dose-dependently during adipose differentiation. The goal of our study was to check if stevioside suppresses the differentiation of pre-adipocytes.
In Figure 2e, C/EBPα expression does not seem to be decreased as it is shown in Figure 2c.
à Thank you for your comment. As you can see in figure 2c, the C/EBPα expression has decreased. This density was measured by Amersham Imager 680 Analysis Software.
For your information, we provide the measured excel results.
|
NC |
Con (0) |
25 |
50 |
100 |
200 |
|
0.003225 |
1.01955 |
0.511045 |
0.51669 |
0.294805 |
0.034639 |
|
0.004068 |
1.000845 |
0.59458 |
0.5083 |
0.20247 |
0.037853 |
|
0.003418 |
1.000622 |
0.620349 |
0.545379 |
0.260126 |
0.046886 |
Phosphorylation of ACC leads to suppression of fatty acid synthesis, and it is consistent with the decreased expression of Srebp-1c and Fas. However, it seems to me that the protein expression levels of ACC are increased by stevioside treatment (Figure 4b). Phosphorylated ACC is also increased, but isn’t it because the ACC expression itself is increased by stevioside? The graph does not seem to represent the images. In addition, in the Discussion section, the authors wrote that stevioside reduced the production of ACC. This is contradictory.
à Thank you for your comment. As you can see in Figure 4b, the expressions of p-ACC and ACC are increased stevioside treatment. However, the ratio of p-ACC/ACC is incurred until 100 μM following stevioside treatment. The fact that stevioside treatment reduces the production of ACC compared to p-ACC led to this outcome. This density was measured by Amersham Imager 680 Analysis Software. For your information, we provide the measured excel results.
|
NC |
Con (0) |
25 |
50 |
100 |
200 |
|
0.470324 |
1.022912 |
1.810431 |
2.205176 |
2.023627 |
0.963761 |
|
0.46784 |
1.01955 |
1.701535 |
1.933523 |
2.012948 |
1.026185 |
|
0.5316249 |
1.0422926 |
1.758827 |
2.1519922 |
2.1201901 |
1.0412498 |
The authors aim to demonstrate that β-oxidation is stimulated by stevioside by assessing the expression of CPT1, SIRT1, and PGC-1a. However, the data does not fully support this conclusion. Specifically, authors need to show the expression and phosphorylation levels of lipolytic proteins such as ATGL and HSL, as well as the expression of fatty acid oxidases. Functional demonstration of increased lipolysis such as measurement of NEFA release into the culture medium would also be needed.
à Thank you for your critical comment. As stated in the title, our study focused on the anti-adipogenic effect and β-oxidation by activating AMPK for this purpose, we measured ACC, p-ACC, CPT1, PGC-1α, FAS, and SIRT1.
Mitochondrial β-oxidation represents a crucial process in energy metabolism and is tightly regulated by interactions between the key enzymes carnitine palmitoyltransferase1 (CPT1) and acetyl-CoA carboxylase (ACC). AMPK is a significant regulator of ACC activities. CPT-1 is a key mitochondrial enzyme in the fatty acid oxidation (FAO) pathway (Moon et al., 2016). Most of the studies indicate that PGC-1α regulates the expression of genes controlling both lipid oxidation and synthesis. PGC-1α upregulates the expression of the mitochondrial FAO pathway (Calvo checking, 2008).
In future research, we will look at fatty acid oxidation about stevioside using hormone-sensitive lipase (HSL), adipose triglyceride lipase (ATGL), and NEFA release measures.
Moon J.S., Nakahira K., Chung K.P., et al., (2016). NOX4-dependent fatty acid oxidation promotes NLRP3 inflammasome activation in macrophages. Nat Med. Sep;22(9):1002-12. doi: 10.1038/nm.4153. Epub 2016 Jul 25.
Calvo, J. A., Daniels, T. G., Wang, X., Paul, A., Lin, J., Spiegelman, B. M., et al. (2008). Muscle-specific expression of PPARgamma coactivator-1alpha improves exercise performance and increases peak oxygen uptake. J. Appl. Physiol. 104, 1304–1312. doi: 10.1152/japplphysiol.01231.2007
In Figure 6, it is shown that adipocyte size is significantly decreased in db/db_SS mice while the weight of epididymal WAT did not change. Does this mean that the adipocyte number was increased in db/db_SS mice? If so, adipocyte differentiation should be increased by stevioside treatment. Does this phenotype reflect the in vitro results in Figure 1?
à Thank you for your query. In figure 6, we found the reduced epididymal adipose tissues by stevioside in db/db mouse. Epididymal fat showed that decreasing tendency but was not significant. However, measuring the epididymal fat area reduced significantly.
Weight gain seems to be mainly associated with cell expansion (hypertrophy) (Spalding et al., 2008). In contrast, weight loss leads to a decreased fat cell size independent of the method employed for weight loss: diet restriction, physical activity, or lifestyle intervention (Murphy et al., 2017).
In addition, this decrease in fat size in vivo reflects the reduction in lipid droplets in vitro data.
Spalding KL, Arner E, Westermark PO, Bernard S, Buchholz BA, Bergmann O, Blomqvist L, et al., (2008) Dynamics of fat cell turnover in humans. Nature 453: 783–787. doi:10.1038/nature06902.
Murphy J, Moullec G, Santosa S. Factors associated with adipocyte size reduction after weight loss interventions for overweight and obesity: a systematic review and meta-regression. (2017) Metabolism 67: 31–40. doi:10.1016/j.metabol.2016.09.009.
Phosphorylation and expression of AMPK may be affected by the food intake. The authors should state the time they sacrificed the mice, and if they were starved for some time prior to the sacrifice, knowing importanceated.
à Thank you for your critical comment. We understand the knowing importance of relevant rack of time so we did our best to keep it relevant. We have included in the materials and methods of the manuscript, ‘All mice were starved for 12 ± 1 h prior to sacrifice.’
In Figure 9, stevioside treatment increased the relative level of p-AMPK even in the presence of dorsomorphin. To my understanding, dorsomorphin inhibits the phosphorylation of AMPK, not the expression. So I wonder why stevioside stimulation still phosphorylates AMPK in the presence of dorsomorphin.
à As you can see in figure 9, dorsomorphin (No. 5) inhibits the phosphorylation of AMPK compared to that without treatment (No. 1). When stevioside was treated (No. 2, 4, 6), the p-AMPK/AMPK was continuously increased. So, we think that stevioside activates AMPK.
In the Discusstitle on nonon te the, authors wrote that stevioside could induce the browning of WAT via AMPK signaling. However, this is just speculation since no experiments had been done to prove this. I think this discussion is not appropriate.
à Thank you for your critical comment. As commented, we revised the sentence. “This indicated that stevioside could possess a synergistic ability to induce the browning of white fat.” “PPARγ is known to play an important role in adipocyte differentiation, maintenance, and function.”

Round 2
Reviewer 2 Report
Regarding Fig. 9, I'm still not convinced.
- It seems that phosphorylation of AMPK is slightly inhibited in dorsomorphin treated samples (#5) compared to control (#1). If so, authors should analyze the statistical significance and show it in the figure.
- Phosphorylation of #6 is significantly higher than that of #5. It means AMPK is phosphorylated even in the presence of dorsomorphin. To my understanding, dorsomorphin inhibits the phosphorylation of AMPK. Then, wouldn't it mean that dorsomorphin is not working? If you want to prove that stevioside stimulation promotes the phosphorylation of AMPK, you have to at least show the statistical significance between #2 and #6.
Author Response
Dear Reviewer 2
Thank you for taking the time to check the results of our study, and we appreciate your comments have improved the manuscript considerably. We hope that the revised manuscript will be published in Cells.
Regarding Fig. 9, I'm still not convinced.
It seems that phosphorylation of AMPK is slightly inhibited in dorsomorphin treated samples (#5) compared to control (#1). If so, authors should analyze the statistical significance and show it in the figure.
à Thank you for your comment. We have revised Figure 9 to show the statistical significance between control (#1) to dorsomorphin-treated samples (#5) (p < 0.05). Kindly refer to the red highlighted portion in the revised Figure 9.
Phosphorylation of #6 is significantly higher than that of #5. It means AMPK is phosphorylated even in the presence of dorsomorphin. To my understanding, dorsomorphin inhibits the phosphorylation of AMPK. Then, wouldn't it mean that dorsomorphin is not working? If you want to prove that stevioside stimulation promotes the phosphorylation of AMPK, you have to at least show the statistical significance between #2 and #6.
à Thank you for your comment. We have revised Figure 9 to show the statistical significance between stevioside treated samples (#2) to dorsomorphin and stevioside treated samples (#6) (p < 0.001). Kindly refer to the purple highlighted portion in the revised Figure 9.

This manuscript is a resubmission of an earlier submission. The following is a list of the peer review reports and author responses from that submission.
Round 1
Reviewer 1 Report
In their manuscript Park et al. study the effect of stevioside on adipocyte differentiation. Stevioside is a sweetener present in stevia. The authors propose that stevioside inhibits adipocyte differentiation in 3T3L1 and in rodents and increase AMPK activation. The activation of AMPK by stevioside has already been published, by different teams.
A - It appears that two exposures of the same Western blot have been used for two different proteins. In fig 6 Western blots for PPARg and C/EBPa appear similar. The feeling is reinforced after looking at the original blots. Not only is the background identical but, although the two proteins have different molecular weights (57 and 43 kda), a superposition of the two blots show that, compared to the molecular weight standards on the left, they migrate exactly at the same height, indicating that these are two different exposures of the same western blot.
B - The authors show that after 3 weeks of stevia gavage the size of adipocytes in db/db mice is decreased. Several crucial information is missing.
- Is the weight of the animals modified by stevia?
- db/db mice are hyperphagic; does stevia decrease food intake?
- Is the weight of the different fat depot modified?
- What about metabolic parameters? Does glycemia, circulating lipids, insulin etc.. of db/db mice are decreased by stevia?
- Does stevia ameliorate insulin and glucose tolerance tests in db/db animals?
- Since AMPK is known to increase brown adipose tissue does stevia increases brown adipose depots?
C- The authors show that stevia inhibits adipocyte differentiation and activates AMPK phosphorylation. This is a correlation but not evidence of causation is provided. Indeed, there is no evidence that the level of activation of AMPK induced by stevioside is sufficient to inhibit adipocyte differentiation. To establish a causal relation, the authors must perform experiments showing that stevia does not inhibit adipocyte differentiation in cells lacking AMPK.
D- Why did the authors choose the db/db model? Although the authors stressed that “Obesity is generally caused by the consumption of foods containing a high proportion of fat and sugar, and contains calories higher than those lost through physical activity” they used db/db mice that have a mutation in the leptin receptor and thus become obese because of hyperphia. A model of high-fat-diet induced-obesity would have been more relevant.
E - The purpose of fig 8 is unclear. The authors show that stevioside increase the effect of very low doses of AICAR (10 microM) on AMPK phosphorylation and slightly increase AMPK phosphorylation in presence of dorsomorphin. How does this help to “ determine whether AMPK plays a crucial role in the anti-adipogenic effects of stevioside in 3T3-L1 adipocytes “ as indicated by the authors? Moreover, how do the authors explain that, in addition to decrease AMPK phosphorylation, dosomorphin decreases sharply AMPK expression as can be observed in the western blot?
F - Why do the author write in the abstract that “ Immunoblot analysis revealed that stevioside upregulated the expression of peroxisome proliferator-activated receptor-gamma (PPARγ) and downregulated the expression of sterol regulatory element-binding protein-1c (SREBP-1c), CCAAT/enhancer-binding protein alpha (C/EBPα), and fatty acid synthase “ but show in the results that all these proteins are downregulated by stevia?
Author Response
Reviewer 1
In their manuscript Park et al. study the effect of stevioside on adipocyte differentiation. Stevioside is a sweetener present in stevia. The authors propose that stevioside inhibits adipocyte differentiation in 3T3L1 and in rodents and increase AMPK activation. The activation of AMPK by stevioside has already been published, by different teams.
A - It appears that two exposures of the same Western blot have been used for two different proteins. In fig 6 Western blots for PPARg and C/EBPa appear similar. The feeling is reinforced after looking at the original blots. Not only is the background identical but, although the two proteins have different molecular weights (57 and 43 kda), a superposition of the two blots show that, compared to the molecular weight standards on the left, they migrate exactly at the same height, indicating that these are two different exposures of the same western blot.
à Thank you for the critical comment, and we are very sorry for the error. As per the reviewer’s suggestion, we’ve changed western blots for PPARγ. There was a mistake in showing the immunoblot analysis of PPARγ in figure 6.
B - The authors show that after 3 weeks of stevia gavage the size of adipocytes in db/db mice is decreased. Several crucial information is missing.
à Thank you for your critical opinion. Now we are preparing two more different papers, so we checked many things about the effect of stevia & stevioside in diabetes. A few months later, we plan to publish all the results.
- Is the weight of the animals modified by stevia?
à Yes, the weight loss by stevia (or stevioside) was confirmed, but it was not significant compared to the control group (db/db mice)
- db/db mice are hyperphagic; does stevia decrease food intake?
à Yes, the food intake of stevioside-treated mice was decreased, and it was significant compared to the control group (db/db mice).
- Is the weight of the different fat depot modified?
à The subcutaneous fat weight of stevioside treated mice was decreased, but it was not significant compared to the control group.
- What about metabolic parameters? Does glycemia, circulating lipids, insulin etc.. of db/db mice are decreased by stevia?
à Yes. Fasting blood glucose levels (FBGLs), serum TG, TC, and serum insulin levels were decreased by stevioside.
- Does stevia ameliorate insulin and glucose tolerance tests in db/db animals?
à Yes, it is. Oral Glucose Tolerance Test (OGTT) & Insulin Tolerance Test (ITT) were decreased by stevioside.
- Since AMPK is known to increase brown adipose tissue, does stevia grows brown adipose depots?
à AMPK is a critical player in regulating energy metabolism, and the broad spectrum of activities of AMPK in lipid makes it a desirable target for drug discovery. The stevioside activates the p-AMPK/AMPK of epididymal in db/db mice, but it does not mean that stevioside made more brown adipose tissues than white tissues. So, we do not know about the brown adipose depots of epididymal in db/db mice. Maybe the following study, we have to check about that.
C- The authors show that stevia inhibits adipocyte differentiation and activates AMPK phosphorylation. This is a correlation but no evidence of causation is provided. Indeed, there is no evidence that the level of activation of AMPK induced by stevioside is sufficient to inhibit adipocyte differentiation. To establish a causal relation, the authors must perform experiments showing that stevia does not inhibit adipocyte differentiation in cells lacking AMPK.
à Thank you for your good opinion. In figure 8, we showed that stevioside activates the p-AMPK/AMPK in Dorsomorphin (Compound C, 10 μM) treatment during the adipocyte differentiation. The dorsomorphin has been used as an inhibitor of AMPK in 3T3-L1 (a murine cell line of adipocytes) cells. During the differentiation, we were co-treated with stevioside with/without the AICAR (or Dorsomorphin) to check that stevioside activates the p-AMPK/AMPK during the adipocyte differentiation.
D- Why did the authors choose the db/db model? Although the authors stressed that “Obesity is generally caused by the consumption of foods containing a high proportion of fat and sugar, and contains calories higher than those lost through physical activity” they used db/db mice that have a mutation in the leptin receptor and thus become obese because of hyperphia. A model of high-fat-diet induced-obesity would have been more relevant.
à Thank you for the critical comment. We used db/db model mice to show that Stevioside has an anti-adipogenic effect by activating AMPK in the epididymal adipose tissues of db/db mice.
We used db/db mice instead of high fat diet-induced obesity mice because it takes too long to induce obesity by a high-fat diet, and recently many studies have widely used db/db model mice as an obesity model. Please refer to the information below.
“The development of high fat diet-induced obesity (19 weeks) in C57 B1/6J mice could be divided into three stages: (1) an early stage in response to a high-fat diet that mice were sensitive to exogenous leptin; (2) a reduced food intake stage when mice had an increase in leptin production and still retained central leptin sensitivity; and (3) an increased food intake stage, accompanied by a reduction of central leptin sensitivity (Lin et al. 2000).”
The genetic leptin-deficient ob/ob mice and the leptin-resistant db/db mice are widely used as animal models to study obesity and related metabolic disorders (Suriano et al., 2021; Brun et al., 2007; Everard et al., 2011; Giesbertz et al., 2015; Geurts et al., 2011). Indeed, the ob/ob mice develop obesity and mild insulin resistance, while the db/db mice develop obesity and diabetes. These differences are not yet fully understood as many mechanistic details associating leptin signaling with the development of an obese and a diabetic state remains poorly investigated.
Suriano, F., Vieira-Silva, S., Falony, G. et al. Novel insights into the genetically obese (ob/ob) and diabetic (db/db) mice: two sides of the same coin. Microbiome 9, 147 (2021).
Brun P, Castagliuolo I, Leo VD, Buda A, Pinzani M, Palù G, et al. Increased intestinal permeability in obese mice: new evidence in the pathogenesis of nonalcoholic steatohepatitis. Am J Physiol Gastrointest Liver Physiol. 2007;292(2): G518–25.
Everard A, Lazarevic V, Derrien M, Girard M, Muccioli GG, Neyrinck AM, et al. Responses of gut microbiota and glucose and lipid metabolism to prebiotics in genetic obese and diet-induced leptin-resistant mice. Diabetes. 2011;60(11):2775–86.
Giesbertz P, Padberg I, Rein D, Ecker J, Höfle AS, Spanier B, et al. Metabolite profiling in plasma and tissues of ob/ob and db/db mice identifies novel markers of obesity and type 2 diabetes. Diabetologia. 2015;58(9):2133–43.
Geurts L, et al. Altered gut microbiota and endocannabinoid system tone in obese and diabetic leptin-resistant mice: impact on apelin regulation in adipose tissue. Front Microbiol.2011;2:149.
E - The purpose of fig 8 is unclear. The authors show that stevioside increases the effect of very low doses of AICAR (10 microM) on AMPK phosphorylation and slightly increases AMPK phosphorylation in the presence of dorsomorphin. How does this help to “determine whether AMPK plays a crucial role in the anti-adipogenic effects of stevioside in 3T3-L1 adipocytes “as indicated by the authors? Moreover, how do the authors explain that, in addition to decrease AMPK phosphorylation, dosomorphin decreases sharply AMPK expression as can be observed in the western blot?
à Figure 8 showed that stevioside activates the p-AMPK/AMPK in Dorsomorphin (Compound C, 10 μM) treatment. The dorsomorphin has been used as an inhibitor of p-AMPK/AMPK in 3T3-L1 (a murine cell line of adipocytes) cells. When dorsomorphine is treated during the differentiation, cells’ p-AMPK/AMPK activity has significantly impaired. Still, the group treated with stevioside shows that the p-AMPK/AMPK activity was increased considerably.
F - Why do the author write in the abstract that “Immunoblot analysis revealed that stevioside upregulated the expression of peroxisome proliferator-activated receptor-gamma (PPARγ) and downregulated the expression of sterol regulatory element-binding protein-1c (SREBP-1c), CCAAT/enhancer-binding protein alpha (C/EBPα), and fatty acid synthase “but show in the results that all these proteins are downregulated by stevia?
à Thank you for the critical comment, and we are very sorry for the error. As per the reviewer’s suggestion, we’ve changed the words in the abstract is below.
“Immunoblot analysis revealed that stevioside upregulated downregulated the expression of peroxisome proliferator-activated receptor-gamma (PPARγ), sterol regulatory element-binding protein-1c (SREBP-1c), CCAAT/enhancer-binding protein alpha (C/EBPα), and fatty acid synthase (FAS).”

Reviewer 2 Report
The manuscript aims to dissect the potential anti-adipogenic role of the biomolecule stevioside in both adipocyte and obese mouse models. The data reveals that the stevioside could suppress adipogenesis while induce fatty acid oxidation through AMPK activation, leading to reduction in lipid storage and adipocyte size in 3T3-L1 adipocytes. In vivo administration of stevioside reduces volume of white adipose tissues in genetic obese mouse model (db/db). The study highlights the involvement of AMPK-regulated biological processes in adipocytes upon stevioside stimulation that may hold potential value for obesity intervention. But the current study is less developed, and the mechanistic part is not fully elucidated, for instance, how stevioside regulates adipogenesis and AMPK is unexplored. Several significant errors and concerns also need to be addressed for future consideration.
Major concerns:
1.Fig.1, for cytotoxicity determination, the control (preadipocytes) wasn’t treated as long as the differentiated cells (3 days vs 7 days). Is stevioside present through the whole differentiation process? How about a short term treatment of stevioside in fully differentiated adipocytes (7 days post initiation)? Would you observe the same changes? This is critical because if stevioside mainly target differentiation (the first 4 days), all the metabolic processes would be undoubtedly altered.
- Fig.2, inconsistent statement. Abstract states upregulation of PPARgamma by stevioside but Fig.2 shows downregulation.
- Fig.3, Sirt1 is not well-recognized as a indicator of mitochondrial oxidation in split of its involvement in PGC1a activation. How about other genes more related to fatty acid oxidation (eg: PPARalpha, Pgc1a, acox1, acads)?
- It is unclear how stevioside regulates adipogenesis and AMPK. Are the changes of AMPK/FAO a consequence of impaired adipogenesis?
- How about glucose/insulin tolerance in stevioside-administrated mice? Are blood glucose/insulin levels changed?
- Fig.5, the quantification of fat tissue is confusing. Did you measure the average size of one adipocyte? That will be more comparable among groups as the area of a staining slide could vary due to other factors (eg: cutting). How about the total fat tissue weight? If the fat is shrinking in treatment group, do you observe an increase fatty acid level in the blood? If not, why?
- Fig.6, sample number should be provided in legends. It seems like the stevioside treated mice have defect in adipogenesis. Do you observe increase in preadipocyte population, as determined by pre-adipocyte marker Pref1?
- Fig.8, Were these experiments performed in differentiating cells (drugs were present in the whole differentiation) or in differentiated cells? How long was the treatment? If this experiment is performed in mature adipocytes, will the adipogenic markers be changed?
Minor concerns:
- Page 3, 2.7 statistical analysis, a typo is found in “Mean +/- SED”.
Author Response
Reviewer 2
The manuscript aims to dissect the potential anti-adipogenic role of the biomolecule stevioside in both adipocyte and obese mouse models. The data reveals that the stevioside could suppress adipogenesis while induce fatty acid oxidation through AMPK activation, leading to reduction in lipid storage and adipocyte size in 3T3-L1 adipocytes. In vivo administration of stevioside reduces volume of white adipose tissues in genetic obese mouse model (db/db). The study highlights the involvement of AMPK-regulated biological processes in adipocytes upon stevioside stimulation that may hold potential value for obesity intervention. But the current study is less developed, and the mechanistic part is not fully elucidated, for instance, how stevioside regulates adipogenesis and AMPK is unexplored. Several significant errors and concerns also need to be addressed for future consideration.
Major concerns:
1.Fig.1, for cytotoxicity determination, the control (preadipocytes) wasn’t treated as long as the differentiated cells (3 days vs 7 days). Is stevioside present through the whole differentiation process? How about a short term treatment of stevioside in fully differentiated adipocytes (7 days post initiation)? Would you observe the same changes? This is critical because if stevioside mainly target differentiation (the first 4 days), all the metabolic processes would be undoubtedly altered.
à Thank you for the critical comment. We changed the media containing stevioside within 48 or 72 hours during differentiation, and after 7 days later, we stopped the differentiation. During cytotoxicity measurements, a high concentration of stevioside (500 μM) was identified for 72 hours, and the actual concentration used in the experiment was 200 μM. As shown in Figure 1a, the stevioside (200 μM) treatment group looks similar to the control group. And only stevioside concentrations were different, and gene expression and protein expression were confirmed after 7 days of differentiation under the same conditions.
- Fig.2, inconsistent statement. Abstract states upregulation of PPARgamma by stevioside but Fig.2 shows downregulation.
à Thank you for the critical comment, and we are very sorry for the error. As per the reviewer’s suggestion, we’ve changed the words in the abstract is below.
“Immunoblot analysis revealed that stevioside upregulated downregulated the expression of peroxisome proliferator-activated receptor-gamma (PPARγ), sterol regulatory element-binding protein-1c (SREBP-1c), CCAAT/enhancer-binding protein alpha (C/EBPα), and fatty acid synthase (FAS).”
- Fig.3, Sirt1 is not well-recognized as an indicator of mitochondrial oxidation in a split of its involvement in PGC1a activation. How about other genes related to fatty acid oxidation (eg: PPARalpha, Pgc1a, acox1, acads)?
à Thank you for the critical comment. Sirt1 is a functioning regulator of PGC-1α that induces a metabolic gene transcription program of mitochondrial fatty acid oxidation. And there are many reports about the Sirt1. Please refer to the paper below.
Gerhart-Hines, Z. et al. Metabolic control of muscle mitochondrial function and fatty acid oxidation through SIRT1/PGC-1α. The EMBO Journal 2007;26: 1913-1923.
Ajay A. et al. SIRT1 Mediates Enhanced Mitochondrial Oxidative Phosphorylation in Chronic Myelogenous Leukemia Stem Cells. Blood 2018; 132 (Supplement 1): 932.
“In vitro and in vivo studies have highlighted a key role for SIRT1 in adipogenesis (Mayoral, R. et al. 2015; Hui, X. et al. 2016; Guarente, L. 2006) and its targets include transcription factors like PPARγ and PGC-1α (Garcia-Jimenez, C. et al. 2016; Mayoral, R. et al. 2015).”
Mayoral, R. et al. Adipocyte SIRT1 knockout promote PPARgamma activity, adipogenesis, and insulin sensitivity in chronic-HFD and obesity. Mol. Metab. 2015; 4, 378–391.
Hui, X. et al. Adipocyte SIRT1 controls systemic insulin sensitivity by modulating macrophages in adipose tissue. EMBO Rep. 2016;18, 645–657.
Guaranteed, L. Sirtuins as potential targets for metabolic syndrome. Nature. 2006; 444, 868–874.
Garcia-Jimenez, C. et al. From obesity to diabetes and cancer: Epidemiological links and role of therapies. Br. J. Cancer. 2016;114, 716–722.
- It is unclear how stevioside regulates adipogenesis and AMPK. Are the changes of AMPK/FAO a consequence of impaired adipogenesis?
à Thank you for the critical comment. In our study, stevioside increased the protein levels of CPT1& Sirt1 and p-AMPK/AMPK & p-ACC/ACC at adipose tissues in vivo. In addition, stevioside has anti-adipogenic effects in the 3T3-LI cell during the differentiation through the AMPK pathway.
AMPK plays a crucial role in regulating adipogenesis (Wu L. et al. 2018), and stevioside exerts an anti-adipogenic effect on adipocyte differentiation by activating the AMPK signaling pathway.
In the case of obesity, AMPK remains inactive due to the availability of excess nutrients and energy sources, therefore an external stimulus would be needed to activate AMPK (Kim J, et al 2016). And much work has been performed to delineate the exogenous activators of AMPK like 5-Amino-4-imidazole carboxamide riboside (AICAR) was the first identified direct activator AMPK in vitro and in vivo.
Therefore, we did not know whether stevioside is performed like an exogenous activator of AMPK but found that stevioside has anti-adipogenic effects through the AMPK pathway. And we need more studies on this.
Wu L, Zhang L, Li B, et al. AMP-Activated Protein Kinase (AMPK) Regulates Energy Metabolism through Modulating Thermogenesis in Adipose Tissue. Front Physiol. 2018;9:122.
Kim J., Yang G., Kim Y., Kim J., Ha J. AMPK activators: mechanisms of action and physiological activities. Exp. Mol. Med. 2016; 48:e224-12.
- How about glucose/insulin tolerance in stevioside-administrated mice? Are blood glucose/insulin levels changed?
à Thank you for your good question. Fasting blood glucose levels (FBGLs), serum TG, TC, and serum insulin levels were decreased by stevioside. In addition, stevioside lowered the Oral Glucose Tolerance Test (OGTT) & Insulin Tolerance Test (ITT) were reduced by stevioside.
Now we are preparing two more different papers, so we checked many things about the effect of stevia & stevioside in diabetes. A few months later, we plan to publish all the results.
- Fig.5, the quantification of fat tissue is confusing. Did you measure the average size of one adipocyte? That will be more comparable among groups as the area of a staining slide could vary due to other factors (e.g.: cutting). How about the total fat tissue weight? If the fat is shrinking in treatment group, do you observe an increase fatty acid level in the blood? If not, why?
à We measure the average size of adipocytes among groups. The weights of subcutaneous fat and epididymal fat of stevioside treated mice were decreased, but it was not significant. The serum TG & TC of stevioside treated mice were significantly reduced compared to those of the control group.
- Fig.6, sample number should be provided in legends. It seems like the stevioside treated mice have defect in adipogenesis. Do you observe increase in preadipocyte population, as determined by pre-adipocyte marker Pref1?
à Thank you for the significant comment. Sample explanations were added in figure 6 & 7 legends like below. Also, the figure was made clearer so that the samples could be distinguished (without numbers).
“C57BL/6; negative control group that received saline, db/db; control group that received saline, db/db_SS; control group that received 40 mg/kg of stevioside.”
We didn’t check the Pref1 marker. We only focused on the adipogenesis of 3T3-L1 cells. In the following study, we need to check the pre-adipocyte marker, Pref1.
- Fig.8, Were these experiments performed in differentiating cells (drugs were present in the whole differentiation) or in differentiated cells? How long was the treatment? If this experiment is performed in mature adipocytes, will the adipogenic markers be changed?
à In figure 8, we performed the 7 days in AICAR (10 μM) or Dorsomorphin (Compound C, 10 μM) treatment with stevioside during the adipocyte differentiation.
Minor concerns:
- Page 3, 2.7 statistical analysis, a typo is found in “Mean +/- SED”.
à Thank you for your comment. We changed the typo to “mean ± standard deviation (SD)”. And also, we have changed all the figure legends like this “All data are presented as mean ± SD”.

Round 2
Reviewer 1 Report
Beside the western blot mistake, no question has been answered.
B- In their answers to the reviewer comments concerning the physiological parameters of obese mice treated with stevia the authors indicate that “they are preparing two more different papers”. Since no new information are provided in this manuscript these data belong in this manuscript.
C- Figure 8 show that dosomorphin is an inhibitor of AMPK. No data on differentiation is provided. So the original comment still stands: there is no data in this manuscript that indicates that stevia inhibit adipocyte differentiation through AMPK activation.
D the answer that “We used db/db mice instead of high fat diet-induced obesity mice because it takes too long to induce obesity by a high-fat diet,” is not satisfactory.
E- I don’t undertand how the observation that dosomorphin does not totally inhibits stevia-induced AMPK activity is relevant to the message of this paper.
Reviewer 2 Report
Thanks for the efforts to revise the manuscript accordingly. Although the revised version successfully addressed some previously raised concerns, remaining questions/errors still need to address for the final version. The mechanisms are still not fully elucidated, whether or not the observation in AMPK/oxidation in treatment group is an indirect outcome from impaired adipogenesis, is unclear.
- Many bar chart (Fig.1b and Fig.2a, Fig.5b) disappear in the PDF file.
- Sirt1 is a master regulator in multiply signaling pathways including PGC-1-mediated mitochondrial oxidation. But changes in Sirt1 expression itself does not necessarily indicate alterations in PGC activity and downstream oxidation. The only direct evidence shown in this manuscript is CPT. A panel of oxidation genes will help to strengthen the conclusion drawn here.
- As the in vivo GTT/ITT data is available, I will recommend to include this to strengthen the significance of this manuscript.
- Fig.5b, as the authors already clarify the determination on average adipocyte size, the corresponding figure legend will also need to revise. Also, tissue weight is important parameter that should be included.
- Mouse sample number is still missing.
- The outcomes incurred from stevioside are likely a combined effect in adipogenesis and oxidation. Unfortunately, the current version could not distinguish the weight and interplay of these two processes. Similarly, the in vivo phenotypes could be attributed by impaired adipogenesis as well, although one can not deny the importance of increase in fatty acid oxidation.